# Coupled Least Squares Support Vector Ensemble Machines

**Dickson Keddy Wornyo [1,2]** and **Xiang-Jun Shen [1,*]**

[1] School of Computer Science and Communication Engineering, JiangSu University, Zhenjiang 212013, China; macdicksons@yahoo.com

[2] School of Computer Science, Datalink Institute, P.O. Box CO 2481 Tema, Ghana

[*] Correspondence: xjshen@ujs.edu.cn; Tel.: +86-511-8878-0371

**Abstract:** The least squares support vector method is a popular data-driven modeling method which shows better performance and has been successfully applied in a wide range of applications. In this paper, we propose a novel coupled least squares support vector ensemble machine (C-LSSVEM). The proposed coupling ensemble helps improve robustness and produce good classification performance than the single model approach. The proposed C-LSSVEM can choose appropriate kernel types and their parameters in a good coupling strategy with a set of classifiers being trained simultaneously. The proposed method can further minimize the total loss of ensembles in kernel space. Thus, we form an ensemble regressor by co-optimizing and weighing base regressors. Experiments conducted on several datasets such as artificial datasets, UCI classification datasets, UCI regression datasets, handwritten digits datasets and NWPU-RESISC45 datasets, indicate that C-LSSVEM performs better in achieving the minimal regression loss and the best classification accuracy relative to selected state-of-the-art regression and classification techniques.

**Keywords:** least squares support vector machines; kernel ensemble; regression; classification

## 1. Introduction

Among the support vector machines (SVM) [1,2], the least squares support vector machine (LSSVM) is considered as a variation of the standard support vector machine (SVM) developed by Suykens et al. [3]. The concept of LSSVM has been successfully applied in many literatures to achieve good results. It is used optimally to control non-linear Karush–Kuhn–Tucker systems for both classification and regression. Also, it has been applied in real-world pattern recognition problems such as image classifications, visual tracking and fault detection techniques, among others [4–7]. LSSVM is computationally based on equality constraints in place of inequality constraints. Also, it produces closed-form solutions by solving linear equation systems instead of solving quadratic programming (QP) problems iteratively in the conventional SVM methods. Thus, training using LSSVM is simpler than that of the SVMs. Furthermore, LSSVM is simple to construct and has the ability to avoid over-fitting to aid in achieving a high generalization performance, making LSSVM popular and widely used. Researchers in recent times have also made some contributions towards the robustness of the LSSVM method. For instance, Lu et al. [8] proposed a robust spatiotemporal LSSVM modeling method for a distributed parameter system (DPS) with disturbances. In this model, a spatial kernel function is firstly constructed in order to describe the nonlinear relation amongst the spatial positions. Liu et al. [9] also proposed a robust WLSSVM-PTS based on weighted LSSVM and penalized trimmed squares to overcome the drawback of achieving robust regression in a noisy environment by adding a weight to each training sample.

Despite the computational advantage and attractive features of LSSVM, it has some drawbacks. For example, parameter selection is vulnerable with respect to various kernel types and their parameters. The selection of kernel functions could be very difficult considering the wide diversity of

kernel functions. Moreover, the optimization of parameters is computationally challenging due to the evaluation demands of some cross-validation procedures [10]. To solve this problem, the ensemble model is used. The ensemble model primarily groups several weak learning techniques together to form a strong learning technique. Some well-known ensemble models are Random Forest (RF) [11,12], Gradient Boosting [13,14] and Tree Regression [15,16].

Motivated by the developments discussed above, we propose a novel coupled least squares support vector ensemble machine (C-LSSVEM). The proposed coupling ensemble helps improve robustness and produce good classification performance than the single model approach. The proposed C-LSSVEM can choose appropriate kernel types and their parameters. Moreover, the proposed method can minimize the total loss of ensembles in kernel space. Thus, we form an ensemble regressor by co-optimizing and weighing base kernel regressors. To improve the robustness of the single model, the coupled idea is used to train our ensemble model simultaneously. It is worth noting that the proposed method is similar to yet different from the existing coupled methods used in the field of facial recognition [17,18], artificial neural network [19,20] and partial least square [21,22]. Furthermore, the proposed C-LSSVEM method is different and improves [23] by integrating the coupling strategy to optimize our base model regressors.

To the best of our knowledge, we are the first to propose a coupled ensemble framework of LSSVM. The main contributions of this paper are as follows:

- The proposed model uses the ensemble model to choose suitable kernel types and their parameters. The proposed method can minimize the total loss of ensembles in kernel space. Thus, we form an ensemble regressor by co-optimizing and weighing base kernel regressors.
- Our proposed method improves training base models in a coupling strategy. This helps the base model generate robustness and better classification performance by compelling each local minimizers together to solve training optimization problem in a coupling way.
- Experiments conducted on artificial datasets, UCI datasets, and handwritten digits datasets indicate that the proposed model effectively performs better in achieving the lowest regression loss and the highest classification accuracy as compared to the state-of-the-art methods. Additionally, we test our model on NWPU-RESISC45 dataset with deep features being trained in AlexNet and VGGNet. This shows superiority performance in feature learning and classification.

The rest of this paper is outlined as follows. Section 2 introduces some related works with respect to the topic under discussion. Section 3 presents the proposed method in details. In Section 4, the experimental results are presented. Section 5 concludes this paper finally.

## 2. Related Work

LSSVM has been addressed in a lot of prior studies. In this section, we introduce some related studies on LSSVM and ensemble models.

### 2.1. Least Squares Support Vector Machine

LSSVM has the same classification principle as SVM, but there are differences in solving the hyperplane. SVM uses quadratic programming to optimize parameter hyperplane while LSSVM transforms the linear programming problem of SVM into constraint conditions. Thus, changes the structure of the loss function, hence greatly reduces the computational effort. LSSVM uses this hyperplane to fit the location of the sample points. LSSVM is generally used for optimal control, classification and regression problems [3,24]. LSSVR is introduced as a regression for LSSVM. The LSSVR technique is to approximate a function by using a given sample of a training data series $\{x_1, y_1\}_{i=1}^N$. The regression function can be formulated as a feature space representation:

$$y = f(X) = w^T \delta(x) + b \tag{1}$$

where the $x \in \Re^d, y \in \Re$ and $\delta(.) : \Re^d \mapsto \Re^h$ is the mapping to the high dimensional feature space. The optimization problem of LSSVM is given as:

$$min J_1(w, b, e) = \frac{1}{2} w^T w + \frac{1}{2} C \sum_{i=1}^{n} e_i^2 \tag{2}$$

Subjected to

$$y_i = w^T \delta(x_i) + b + e_i, i = 1, 2, ..., n \tag{3}$$

Research has recently been made to the contributions of the LSSVM method. For example, Zheng et al. [25] proposed a novel model that combines wavelet technique integrated LSSVM with improved PSO for forecasting of dissolved gases in oil-immersed transformers. Wen et al. [26] on the other hand also presented a different method which integrates machine learning and complexity theory to assess node relevance in complex network relying on LSSVMs techniques with experimental outcome showing the accuracy and efficacy of their method.

*2.2. Ensemble Regression*

Ensemble learning is a kind of machine learning paradigm in which multiple models, such as decision trees, neural networks and SVM, are combined together to solve a particular problem [27]. Typical ensemble methods include Adaboost [28], random forests [29] and gradient boosted machines [30]. All these methods encourage diversity of the base learners to some extent to compensate individual errors and reach a better-expected performance.

Adaboost is a common ensemble and iterative algorithm [31] that allows a new classifier to be generated from the training dataset in each iteration; it further classifies all samples to assess the importance of each sample. The weight of the wrongly classified samples will be higher in the next training. The whole process will not end until the error rate is small enough or up to a certain iteration number. Moghimi et al. [32] proposed a vehicle detection technology which aims to locate and show the vehicle size in digital images based on the boosting technique by Viola Jones. Their experimental results showed that the accuracy, completeness, and quality of the proposed vehicle detection method are better than previous techniques. Yin et al. [33] proposed a new method of video text localization based on Adaboost. The experimental results showed that their method does not only achieve a good effect on the text localization in video images with a text of various fonts, sizes and colors but also can realize rapidly and accurately these requirements to meet the video text localization.

The random forest, proposed by Breiman [29] is an ensemble approach that can also be thought of as a form of the nearest neighbor predictor. It is an algorithm that uses multiple trees to train and predict a sample. Melville et al. [34] presented a random forest classification approach for identifying and mapping three types of lowland natives grassland communities found in the Tasmania midlands region. The results of this study indicated that remote sensing is a viable method for the identification of lowland native grassland communities in the Tasmanian Midlands, and that repeat classification and statistical significance testing can be used to identify optimal datasets for vegetation community mapping. Jog et al. [35] presented a supervised random forest image synthesis approach called RELICA, that learns a non-linear regression to predict the intensities of alternate tissue contracts given specific input tissue contracts.

Gradient boosting is an ensemble technique in which the predictors are not made independently, but sequentially. Gradient boosting is one of the most powerful techniques for building predictive models. Li et al. [36] used an extreme gradient boosting regression tree model to analyze twitter signals as a medium for user sentiment to predict the price fluctuations of a small-cap alternative cryptocurrency called (ZClassic). Their model is the first academic proof of concept that social media platforms such as twitter can serve as a powerful social signal for predicting price movements in the highly speculative alternative cryptocurrency or "alt-coin" market. Touzani et al. [37] more recently

presented an energy consumption baseline modeling method based on a gradient boosting machine to assess the performance of testing procedures used on a large dataset of 410 commercial buildings. The results showed that using the gradient boosting machine model improved the R-squared prediction accuracy and the CV(RMSE) in more than 80 percent of the cases when compared to an industry best practice model that is based on piecewise linear regression, and to a random forest algorithm.

## 3. The Proposed Method

In this section, we explore the intricacy of the novel coupled least squares support vector ensemble machine (C-LSSVEM). The following subsections talks about kernel theory and the proposed model respectively

### 3.1. Kernel Theory

Kernel methods map the data into a high dimensional feature space, where each coordinate corresponds to one feature of the data items. In that space, a variety of methods can be used to find relations in the data. Since the mapping can be quite general (e.g., not necessarily linear), the relations found in this way are explicitly general. Kernels are proposed as a result of varied situational and application differences. A Mercer Kernel function $K : X \times X \mapsto \Re$ is said to be symmetrically continuous and positive semidefinite. Thus, for any finite set of distinct points $\{x_1, x_2, ..., x_N\} \in X$, the matrix $\{(x_i, x_j)\}_{i,j=1}^{N}$ is positive semidefinite.

The basic features of a kernel function are derived from Mercer's theorem [38]. Applicable kernel functions must satisfy Mercer's conditions. This study uses the radial function (RBF), the gaussian function and the polynomial function as kernel functions as shown below:

- The Polynomial kernel

$$k(x_i, x_j) = (a x_i^T x_j + b)^c \tag{4}$$

- The RBF kernel (Radial Basis Function)

$$k(x_i, x_j) = exp(-\frac{\|x_i - x_j\|}{\mu}) \tag{5}$$

- The Gaussian kernel

$$k(x_i, x_j) = exp(-\frac{\|x_i - x_j\|^2}{2\sigma^2}) \tag{6}$$

where $a, b, c, \mu, \sigma \in R$. $a, b$ and $c$ are kernel parameters used in the experiment, $\mu$ and $\sigma$ are parameters frequently used by kernels in practice due to its capacity to generate nonparametric classification functions. $x_i - x_j$ represents feature vectors in input space. While **K** denotes a Gram matrix obtained according to samples. Which is a symmetric and semi-positive definite matrix given as follows:

$$\mathbf{K} = \begin{pmatrix} k(x_1, x_1) & k(x_1, x_2) & \cdots & k(x_1, x_N) \\ k(x_2, x_1) & k(x_2, x_2) & \cdots & k(x_2, x_N) \\ \vdots & \vdots & \ddots & \vdots \\ k(x_N, x_1) & k(x_N, x_2) & \cdots & k(x_N, x_N) \end{pmatrix} \tag{7}$$

Given a set of labeled examples $(x_i, y_i), i = 1, ..., N$, the standard framework estimates an unknown function by minimizing:

$$f^* = \arg \min \sum_{i=1}^{N} E(y_m, f(x_m)) + \lambda \|f\|_k^2 \tag{8}$$

where $E : \Re \times \Re \mapsto [0, \infty]$ is a loss function, such as squared loss $(y_i - f(x_i))^2$ for hinge loss or regularized least square loss function $\max[0, 1 - y_i f(x_i)]$ for SVM. $\lambda ||f||_k^2$ is considered as a smooth condition on likely solutions and the lambda is a positive parameter to trade off the balance. Moreover, the classical representation theorem states the solution to minimizing problems that exist can be written as:

$$f(x) = \sum_{j=1}^{N} a_j k(x, x_j) \tag{9}$$

Hence, the difficulty is reduced to enhancing over the finite dimensional space or coefficients $a_j$, which is the algorithmic basis for SVM, regularized least squares, and other regression methods.

### 3.2. Coupled Least Squares Support Vector Ensemble Machine (C-LSSVEM)

In this subsection, we introduce our coupled least squares support vector ensemble method. Diverse kernel models and their parameters are utilized to construct base regressors. The proposed kernel ensemble method is presented as follows.

Diverse kernels are archived according to data samples. Supposing a training set $X$ with regression result ($X = \{(x_1, y_1), ..., (x_N, y_N)\}$) and a testing set $X_t$ without regression result ($X_t = \{(x_1, ..., x_{N_t})\}$) where $x_n (x_n \in R^d, n = 1, ..., N)$ expresses a training sample, $y_n$ is the real value of $x_n$, and $x_t (x_t \in R^d, t = 1, ..., N_t)$ expresses a testing sample. $N$ is the number of training samples and $N_t$ is the number of testing samples. The base kernel regressor is built as a kernel regressor.

On the other hand, diverse kernel types and their parameters selection result in various regression results. So as to get a superior regression ensemble model, base kernel regressors are consolidated in our coupled least squares support vector ensemble framework. Those base regressors are coupled and weighted in the following part. To make simpler the whole model, we present a new variable $e_{in}$, which equals to $([(K_{i+1}\alpha_{i+1} + b_{i+1}1_{N*1})_{n \times 1} - (K_i\alpha_i + b_i 1_{N*1})_{n \times 1}])$. The proposed coupled least squares support vector ensemble machine (C-LSSVEM) model is as follows:

$$\underset{e_i, b_i, w, \alpha_i}{argmin} \sum_{i=1}^{L} w_i \left( \sum_{n=1}^{N} ||(K_i\alpha_i)_n + b_i - y_n||_2^2 \right) + e_{in}^2 + \lambda \alpha_i^T K_i \alpha_i) \tag{10}$$
$$s.t. 1^T w = 1, e_{in} = [(K_{i+1}\alpha_{i+1} + b_{i+1}1_{N*1})_n - (K_i\alpha_i + b_i 1_{N*1})_n]$$

where $L$ is the number of base regressors. $w$ denotes a weight vector of individual base kernel regression model and $w = [w_1, ..., w_L]^T$. $K_i$ is the $i$-th base gram matrix and $e_{in}$ is the coupling error between the $(i + 1)$-th base regressor and the $i$-th base regressor. $\alpha_i$ is $N \times 1$ weight column vector, which is identified to the weight of every training data sample in $K_i$. $b_i$ is the bias item for the $i$-th base regressor.

Equation (10) can be transformed into Equation (11) by adding Lagrangian multiplier $\beta$:

$$\underset{e_i, b_i, w, \alpha_i}{argmin} \sum_{i=1}^{L} w_i (\alpha_i^T K_i^T \alpha_i + 2\alpha_i^T K_i^T b_i 1_{N*1} - 2\alpha_i^T K_i^T y$$
$$+ b_i^2 N - 2b_i 1_{N*1}^T y + y^T y + \lambda \alpha_i^T K_i \alpha_i \tag{11}$$
$$+ e_i^T e_i + 2\beta_i^T [e_i - (k_{i+1}\alpha_{i+1} + b_{i+1}1_{N*1})$$
$$+ (K_i\alpha_i + b_i 1_{N*1})])$$

Derivatives are taken of Equation (11) with respect to $b_i$, $e_i$, $\boldsymbol{\alpha}_i$, and $\beta_i$ obtain as follows, whiles we set them to zero (0):

To derive $e_i$

$$e_i + \beta_i = 0 \tag{12}$$

To derive $\beta_i$

$$e_i = (K_{i+1}\alpha_{i+1} + b_{i+1}1_{N*1}) - (K_i\alpha_i + b_i1_{N*1})\tag{13}$$

To derive $\boldsymbol{\alpha}_i$

$$\begin{aligned}&w_i(K_iK_i\alpha_i + b_iK_i1_{N*1} - K_iy + \lambda K_i\alpha_i + K_i\beta_i)\\&\quad - w_{i-1}K_i\beta_{i-1} = 0\end{aligned}\tag{14}$$

To derive $b_i$

$$\begin{aligned}&w_i[\alpha_i^TK_i1_{N*1} + b_iN - 1_{N*1}y + \beta_i^T * 1_{N*1}]\\&\quad - w_{i-1}[\beta_{i-1}^T * 1_{N*1}] = 0\end{aligned}\tag{15}$$

From the above, we then substitute $\beta_i$ and $e_i$ into Equations (14) and (15) and obtain the following equations:

$$\begin{aligned}\alpha_i =& (2w_iK_i + w_{i-1}K_i + \lambda w_iI)^{-1}(w_iy + w_iK_{i+1}\alpha_{i+1}\\&+ w_{i-1}K_{i-1}\alpha_{i-1} - 2w_ib_i1_{N*1} - w_ib_{i+1}1_{N*1}\\&+ w_{i-1}b_{i-1}1_{N*1} - w_{i-1}b_i1_{N*1})\end{aligned}\tag{16}$$

$$\begin{aligned}b_i =& \frac{1}{(2w_i + w_{i-1})N}(w_ib_{i+1}N + w_{i-1}b_{i-1}N\\&- 2w_i\alpha_i^TK_i1_{N*1}y + w_i\alpha_i^TK_{i+1}1_{N*1}\\&+ w_{i-1}\alpha_{i-1}^TK_{i-1}1_{N*1} - w_{i-1}\alpha_i^TK_i1_{N*1})\end{aligned}\tag{17}$$

Since our approach aims to select suitable kernel types and parameters in individual kernel regressors and also to obtain an optimal weight vector of base regressors, we aimed at minimizing the loss for determining the performance of the base kernel regression models.

Consider $W_i$ to be $W_i^r$ (where $r$ speaks to the control parameter for the weights of multiple features) in light of the fact that linear programming accomplishes its ideal solution at the extreme ends. In this way, either $W_i = 0$ or $W_i = 1$. This implies there will be one kernel chosen in opposing to our goal of discovering the rich complementation of multiple kernels. At the point $r = 1$, just a single kernel will be chosen in the ideal result, which is undesirable, yet on the off chance that $r > 1$ the ideal result is based on multi-kernel adjusting. The value of r is man-made to obtain appropriate w. We can further derive that:

$$w_i = \frac{\left(\frac{1}{\zeta_i}\right)^{r-1}}{\sum\limits_{i=1}^{L}\left(\frac{1}{\zeta_i}\right)^{r-1}}\tag{18}$$

where $\zeta_i = \|K_i\alpha_i + b_i - y\|_2^2 + \lambda\alpha_i^TK_i\alpha_i$ denotes the loss of each kernel. As per Equation (18), the ideal weight of the ensemble method can be achieved, where r is a parameter to get suitable w. We achieve an ensemble regression model by consolidating the different base kernel models linearly. The proposed kernel ensemble regressor is constructed using Formula (19).

$$f(x_t) = \sum_{i=1}^{L} W_i(\sum_{j=1}^{N} K_i(x_j, x_t)\alpha_{i,j} + b_i)\tag{19}$$

The C-LSSVEM method summarized in Algorithm 1.

---

**Algorithm 1** The proposed C-LSSVEM method.

---

1: **Input:** Training set $X$, $L$
2: **Output:** $f(x_t)$
3: **Parameters:** $\alpha$,b,c,$\lambda$,r
4: **Initialize:**

　-Set $w = (\frac{1}{L}, \frac{1}{L}, ..., \frac{1}{L})^T$

　-Constract gram matrix K
5: **while** not converged **do**

6:　　obtain $f(x_t)$ as in Equation (19)
7:　　Update $e$ through Equation (13)
8:　　Update $\alpha$ through Equation (16)
9:　　Update $b$ through Equation (17)
10:　　Update $w$ through Equation (18)
11:　　Compute loss through Equation (10)
12: **end while**

---

## 4. Experiment Result

This section demonstrates the generalization performance advantage of the coupled ensemble multiple kernel based method in our proposed model (C-LSSVEM) over other regression methods, for example, ridge regression (RR), support vector regression (SVR), random forest (RF), gradient boosting regression (GBR), decision tree regression (DTR) and extreme gradient boosting (XGBoost) [39]. To validate the performance of our proposed method, artificial dataset, UC Irvine (UCI) regression and UC Irvine (UCI) classification datasets are used. The details of the experimental settings and results on different datasets are discussed in the following subsection.

### 4.1. Experimental Settings

The experiments are conducted with training (i.e., 2/3) and testing (i.e., 1/3) data from each dataset. It is worth noting that the training and testing data of each dataset are randomly selected. The experimental results are performed 10 times on each dataset.

A demonstration of how several LSSVM models in an ensemble are coupled is discussed in the proposed method. In Equation (4), a single polynomial kernel method is considerably used as the elementary method of an ensemble for all the diverse datasets. This method comprises of three parameters (i.e., a, b, and c). Also, the different values of the parameters yield different effects with respect to the experimental results. Specifically, we set parameters a, b, and c as $a \in \{1 * 1e - 6, 1 * 1e - 5, \cdots, 1000\}$, $b \in \{1 * 1e - 6, 1 * 1e - 5, \cdots, 1000\}$ and $c \in \{1, 2, 3, 4, 5\}$ respectively. The parameter L in Equation (10) demonstrates the number of the base polynomial kernel models. Considering the generalization ability of an ensemble regressor, it is always expedient to have enough base models. Nevertheless, extreme availability of base models possibly will result in a worse generality capacity of an ensemble regressor, which yields a poor classification accuracy level. Therefore, a careful selection of L is given as $L \in \{10, 20, 50, 100, 150\}$ in our experimentations. Moreover, 20 blends among three parameters (i.e., a, b, and c) are selected.

The parameter in Equation (10) is the parameter to smooth the base regressor. The parameter r in Equation (18) is the control parameter for adjusting the weights of multiple base models. The values of $\lambda$ and $r$ are respectively set as 0.1 and 2, in all the experiments.

### 4.2. Experimental Results

In this section, we discuss the overall performance of the proposed C-LSSVEM method with all the relative methods on diverse datasets under the: artificial dataset, UCI regression and UCI classification datasets. The outcomes are recorded in Tables 1–3 with the highest performance results on each dataset highlighted in bolded textual style.

**Table 1.** MSE results (Average ± Std) of different methods on eight UCI datasets.

| Dataset / Method | RR | SVR | RF | GBR | DTR | XGBOOST | C-LSSVEM |
|---|---|---|---|---|---|---|---|
| Abalone | 6.865 ± 0.314 | 7.675 ± 0.128 | 6.223 ± 0.176 | 6.478 ± 0.187 | 6.989 ± 0.135 | 7.103 ± 0.175 | **5.822 ± 0.125** |
| RedWine | 0.522 ± 0.020 | 0.597 ± 0.039 | 0.515 ± 0.009 | 0.516 ± 0.016 | 0.585 ± 0.034 | 0.599 ± 0.065 | **0.401 ± 0.008** |
| Housing | 24.245 ± 3.015 | 33.230 ± 6.030 | 15.304 ± 2.019 | 17.302 ± 0.025 | 20.308 ± 0.029 | 18.672 ± 3.018 | **13.319 ± 0.007** |
| Concrete | 103.857 ± 7.298 | 278.723 ± 11.658 | 43.145 ± 7.095 | 52.543 ± 8.609 | 74.564 ± 7.232 | 54.235 ± 7.342 | **40.123 ± 7.243** |
| Mg | 0.022 ± 0.001 | 0.019 ± 0.001 | 0.018 ± 0.001 | 0.016 ± 0.001 | 0.019 ± 0.001 | 0.017 ± 0.001 | **0.015 ± 0.009** |
| Mpg | 14.902 ± 1.200 | 81.674 ± 3.123 | 10.889 ± 1.725 | 12.565 ± 1.155 | 15.678 ± 1.182 | 13.735 ± 2.016 | **10.143 ± 0.851** |
| Space | 0.023 ± 0.001 | 0.045 ± 0.003 | 0.022 ± 0.001 | 0.023 ± 0.002 | 0.023 ± 0.001 | 0.023 ± 0.001 | **0.022 ± 0.001** |
| Bodyfat | 2.826 ± 0.974 | 45.785 ± 11.099 | 5.673 ± 1.446 | 3.764 ± 0.988 | 3.645 ± 0.978 | 3.989 ± 0.936 | **1.999 ± 0.935** |

**Table 2.** MAE results (Average ± Std) of different methods on eight UCI datasets.

| Dataset / Method | RR | SVR | RF | GBR | DTR | XGBOOST | C-LSSVEM |
|---|---|---|---|---|---|---|---|
| Abalone | 1.723 ± 0.061 | 1.787 ± 0.075 | 1.678 ± 0.052 | 1.753 ± 0.060 | 1.798 ± 0.078 | 1.760 ± 0.082 | **1.658 ± 0.568** |
| RedWine | 0.685 ± 0.013 | 0.764 ± 0.029 | 0.687 ± 0.011 | 0.689 ± 0.012 | 0.732 ± 0.015 | 0.069 ± 0.013 | **0.678 ± 0.010** |
| Housing | 3.504 ± 0.008 | 3.821 ± 0.012 | 2.707 ± 0.008 | 3.104 ± 0.009 | 4.108 ± 0.010 | 3.467 ± 0.010 | **2.101 ± 0.004** |
| Concrete | 10.245 ± 0.198 | 15.673 ± 0.147 | 6.850 ± 0.183 | 7.859 ± 0.918 | 8.467 ± 0.182 | 7.997 ± 0.989 | **7.074 ± 0.218** |
| Mg | 0.128 ± 0.005 | 0.117 ± 0.005 | 0.099 ± 0.005 | 0.100 ± 0.004 | 0.102 ± 0.003 | 0.138 ± 0.007 | **0.090 ± 0.002** |
| Mpg | 3.678 ± 0.150 | 8.737 ± 0.221 | 3.076 ± 0.162 | 3.328 ± 0.140 | 3.598 ± 0.230 | 3.700 ± 0.151 | **3.100 ± 0.030** |
| Space | 0.109 ± 0.001 | 1.622 ± 0.001 | 0.108 ± 0.002 | 0.107 ± 0.005 | 0.107 ± 0.004 | 0.110 ± 0.006 | **0.103 ± 0.001** |
| Bodyfat | 13.253 ± 1.925 | 186.456 ± 20.752 | 57.678 ± 4.967 | 32.364 ± 1.934 | 22.356 ± 2.672 | 33.024 ± 2.000 | **12.873 ± 1.906** |

**Table 3.** Classification accuracy results (%) (Average ± Std) of different methods on five UCI datasets.

| Dataset / Method | RR | SVR | RF | $KNORA-E$ | OLA | C-LSSVEM |
|---|---|---|---|---|---|---|
| Breast-cancer | 97.701 ± 1.365 | 98.825 ± 1.104 | 98.580 ± 1.141 | 95.948 ± 0.800 | 98.367 ± 1.219 | **99.924 ± 0.471** |
| Pima | 79.198 ± 1.820 | 75.466 ± 1.436 | 77.175 ± 1.472 | 76.453 ± 1.865 | 76.895 ± 2.598 | **81.933 ± 0.830** |
| Sonar | 78.323 ± 3.786 | 79.213 ± 4.438 | 80.357 ± 3.678 | 79.898 ± 2.986 | 76.587 ± 3.354 | **81.898 ± 2.673** |
| Australian | 89.566 ± 1.679 | 56.913 ± 4.894 | 86.430 ± 2.166 | 88.980 ± 1.751 | 87.140 ± 1.784 | **89.886 ± 1.946** |
| German | 78.783 ± 1.455 | 72.876 ± 0.018 | 76.784 ± 1.211 | 73.874 ± 2.346 | 76.181 ± 3.536 | **79.783 ± 1.253** |

### 4.2.1. Artificial Dataset

The utilization of artificial dataset is purposefully used to illustrate the performances of the proposed method and comparative methods visually. Ideally, the sampling of the input space is not practical in most cases. So, we utilized it only for demonstration purposes to visualize the regression effects in Figure 1.

Fifty data points are produced from the scalar function corrupted by an observation noise. We made use of the model shown below:

$$y = \frac{sin(\pi x)}{\pi x} + 0.1x + 0.05\eta \tag{20}$$

We visualize the regression effects on five regression methods, namely, SVR, RF, GBR, DTR and C-LSSVEM. We considered the data points to be consistently spread over the *x*-axis on all the methods as shown in the following experimental graph, where $\eta$ denotes noise.

From Figure 1 the fitting of our proposed C-LSSVEM method outperforms the other comparative method in this experiment. From all indications, the loss of the proposed method is the least among the regression methods indicated above. Similar to the SVR method is the GBR and DTR methods where good fitting is achieved by both methods on the *x*-axis however poorly fit midway of the *x*-axis. Our proposed method in Figure 1e shows an excellent regression performance to the comparative methods due to the coupling benefits of both ensemble and kernel methods.

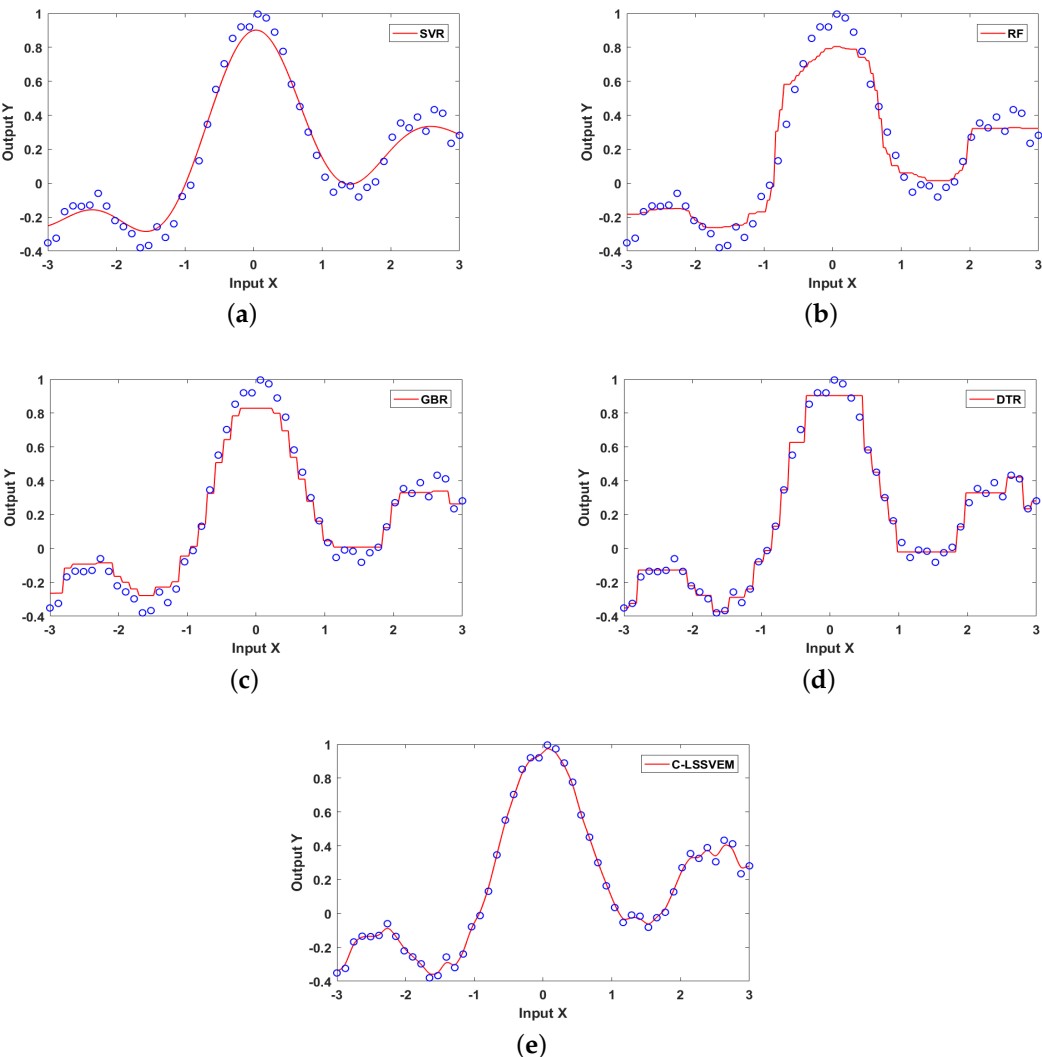

**Figure 1.** The fitting of five different methods on artificial dataset. (**a**) SVR; (**b**) RF; (**c**) GBR; (**d**) DTR; (**e**) C-LSSVEM.

### 4.2.2. UCI-Datasets

(1) Regression

Quite a lot of regression datasets with diverse features have been tested so as to authenticate the performance of our proposed model. We selected eight benchmark publicly available datasets for the evaluation of our performance of our technique from the UCI repository, namely Abalone, Bodyfat, Concrete, Mg, Mpg, RedWine, Space, and Housing. The detailed summary about the UCI datasets used is presented in Table 4 [40]. The standards applied is Mean Absolute Error (MAE) and Mean Square Error (MSE). They are given as:

$$MAE = \frac{1}{N_t} \sum_{i=1}^{N_t} |f(x_i) - y_i| \tag{21}$$

$$MSE = \frac{1}{N_t} \sum_{i=1}^{N_t} (f(x_i) - y_i)^2 \tag{22}$$

where $f(x_i)$ and $y_i$ are the real output and the model output respectively, and $N_t$ is the number of samples. The mean and variance of MSE and MAE are used to evaluate the performance of the proposed method.

**Table 4.** Descriptions of the UCI datasets.

| Dataset | Samples | Attributes |
|---------|---------|------------|
| Abalone | 4177 | 8 |
| RedWine | 1599 | 11 |
| Housing | 506 | 13 |
| Concrete | 1030 | 8 |
| Mg | 1385 | 6 |
| Mpg | 392 | 7 |
| Space | 3107 | 6 |
| Bodyfat | 252 | 14 |

Table 1 presents the average MSE with corresponding standard deviations after running each method ten times. It is evident from the results that our approach performed much better than all the comparative methods. Taking for example the Abalone dataset, our approaches performance superseded RR by 1.042, SVR by 1.853, RF by 0.401, GBR by 0.656, DTR by 1.167 and XGBoost by 1.28. On the Concrete dataset, most of the approaches performed relatively poorly. Our approach however outperformed RF which is the next best performing method on this dataset by 3.022, and also outperformed the worst method which is SVR by 238.6. Taking the standard deviations of all approaches into perspective indicates also that our approach has the best stability, as it consistently records the lowest deviations. The lower MSE and standard deviation values indicates that our proposed approach can better handle non-linear datasets with kernel methods and obtains stable regression performance as a result of the coupling of ensemble methods.

We demonstrate further the merits of coupling kernel and ensemble methods using MAE as a performance measure. Table 2 illustrates the obtained results of the six regression approaches on our selected benchmark datasets. Our proposed C-LSSVEM again consistently outperforms all comparing methods. It records an optimal result of 1.6580 on the Abalone dataset, leading the RF approach by 0.0206. It is also observed that our C-LSSVEM approach attains the best performance of 12.873 and 7.074 respectively on the Bodyfat and Concrete datasets, whiles SVR achieves the poorest performance of 15.673 on the concrete dataset.

Using box diagrams of MSE and MAE on the WhiteWine dataset, which is a regression dataset of 4898 cases in 11 features, which was collected by variants of the Portuguese "Vinho Verde" wine. We again show the preeminence of our approach with the other comparing methods in Figures 2 and 3.

SVR and RR respectively recorded the highest value MSE and MAE amongst all the comparing regression methods from the results in Figures 2 and 3. This is a result of bad parameter selection of SVR and RR. It is observed from the same figures that our C-LSSVEM has the least MSE and MAE values amongst all the comparing methods, making it invariably the best performing method. RF performed quite well on the WhiteWine dataset for both MAE and MSE.

From the above discussions, we deduce that our proposed C-LSSVEM approach has good performance on all the UCI datasets chosen for our experiments. It is also able to select suitable kernels with corresponding parameters that enhances significantly the performance of the regression.

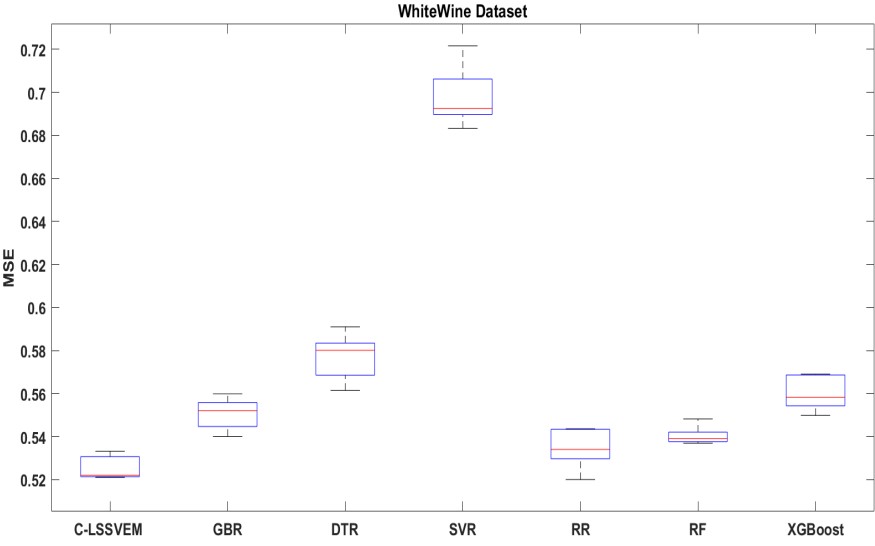

**Figure 2.** The MSE of six regression methods on WhiteWine dataset.

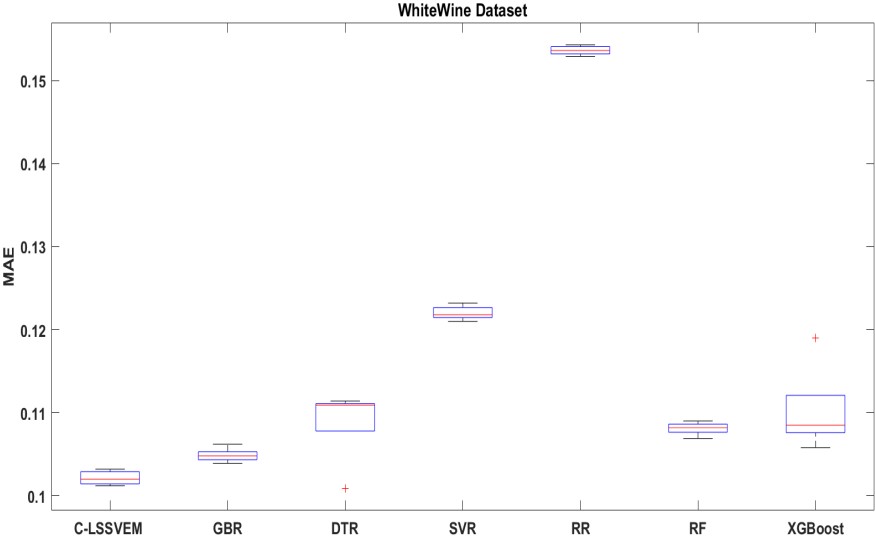

**Figure 3.** The MAE box plot of six regression methods on WhiteWine dataset.

(2) Classification

Even though all the models discussed in the previous sections are for regression tasks, we additionally apply those techniques for classification task to additionally confirm the performance of our proposed C-LSSVEM. Five open datasets from UCI (http://archive.ics.uci.edu/ml/index.php), namely, Breast-cancer, Pima, Sonar, Australian and German are used. Also, we use five comparison algorithms to compare with our method, which includes RR, SVR, RF, K-Nearest Oracles Eliminate (KNORA-E) [41] and Overall Local Accuracy (OLA) [42]. The last two comparison methods are the state-of-the-art techniques for dynamic classifier and ensemble selection in DESlib. The essential information on these datasets is shown in Table 5.

Table 3 summarizes the results of average classification accuracies and corresponding standard deviations of the various comparing methods on selected UCI datasets to further ascertain the efficacy of our approach. The proposed C-LSSVEM obtains higher performance compared to the other approaches under review with respect to all the datasets used in the experiment. For instance on

the Breast-cancer dataset, C-LSSVEM recorded a mean accuracy of 99.9246 with SVR and OLA lagging behind our approach by 1.0992 and 1.5568, respectively. KNORA-E attains the worse performance lagging behind our approach by 3.9757. Analysis on the standard deviations recorded by the various approaches confirms that the proposed C-LSSVEM is more stable than the other baseline approaches since it records the lowest values relative to the comparative models.

**Table 5.** Description information for the five datasets.

| Dataset | Samples | Attributes |
| --- | --- | --- |
| Breast-cancer | 683 | 10 |
| Pima | 768 | 8 |
| Sonar | 208 | 60 |
| Australian | 690 | 14 |
| German | 1000 | 24 |

Additionally, we compare the performance of different comparative approaches. The box plot on two datasets (German and Pima) as illustrated in Figure 4 is used.

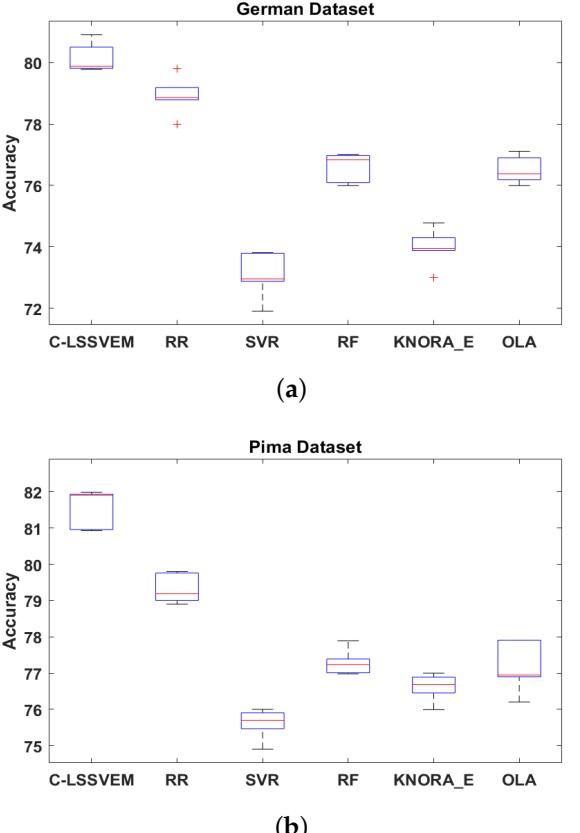

**Figure 4.** The accuracy of six regression methods on two UCI dataset. (**a**) German (**b**) Pima.

Using box plot, we illustrate in Figure 4, the performance of all models on the German and Pima datasets. As asserted in our earlier discussions, Figure 4 further confirms the superiority of our method in classification accuracy with C-LSSVEM recording the highest position amongst the comparative models on the two datasets. The relative narrow shape of the boxes also confirms the stability of our model relative to the other approaches. In both the German and Pima datasets, the RR approach which is a classical method, performed second best to C-LSSVEM, indicating the relevance of our approach which outperforms all the comparative methods in our study.

### 4.2.3. Handwritten Digits-Datasets

In this section, MNIST (http://www.cad.zju.edu.cn/home/dengcai/Data/data.html), USPS (http://www.cad.zju.edu.cn/home/dengcai/Data/data.html) which are handwritten datasets are used in our experiment to perform classification task on five regression methods namelyin our experiments to perform the classification task on six regression methods, namely Adaboost (AB) [43], RR, RF, Simple Vote Rule [44], QFWEC [45] and C-LSSVEM. The detailed descriptions are shown in Table 6 and their accuracy can be seen in Table 7.

**Table 6.** Descriptions information for handwritten digits datasets.

| Dataset | Data Points | No. of Features |
|---------|-------------|-----------------|
| MNIST | 4000 | 784 |
| USPS | 9298 | 49 |

**Table 7.** Classification accuracy result (%) (Average ± Std) of different methods on Handwritten digits datasets.

| Method<br>Dataset | Adaboost | RR | RF | Simple Vote Rule | QFWEC | C-LSSVEM |
|-------------------|----------|-----|-----|------------------|-------|----------|
| MNIST | 74.2345 ± 1.0456 | 69.7635 ± 1.6263 | 88.1793 ± 0.8792 | 87.9267 ± 0.9782 | 83.6671 ± 0.8679 | **93.8726 ± 0.3979** |
| USPS | 82.6728 ± 0.8962 | 89.1782 ± 0.9286 | 93.8369 ± 0.6284 | 93.9872 ± 0.8625 | 94.0432 ± 0.7237 | **94.0768 ± 0.5994** |

Comparing the results of C-LSSVEM from Table 7 to the other methods, the highest mean accuracy performances recorded all belong to our C-LSSVEM method. For instance, 93.8726 ± 0.3979 and 94.0768 ± 0.5994 as recorded for MNIST and USPS, respectively. On the MNIST dataset, it outperformed Adaboost by 19.6381, RR by 24.0191, RF by 5.6933, SVR by 5.9459, and QFWEC by 10.2055. The QFWEC method has high mean accuracy on the USPS dataset and second only to our approach. Although, the result of QFWEC is not satisfactory on the MNIST dataset. In addition, the Simple Vote Rule and RF are optimistic on two datasets. From Table 7, the result of Adaboost is worse on the USPS dataset. The standard deviations values of Table 7 implies our C-LSSVEM method is the most stable amongst the comparative methods as it records the least values in our experiments. From our experimental results on USPS and MNIST datasets, our proposed approach has a good effect on the handwriting field.

### 4.2.4. NWPU-RESISC45 Dataset

In this subsection, we test our model in a large dataset with features learned from deep networks. Deep learning can learn high-level features in data by using structures composed of multiple non-linear transformations. In view of this, we test our model on deep features, which are trained from two kinds of deep learning-based CNN features: AlexNet [46] and VGGNet [47] for its superiority performance in feature learning and classification. The details of these models are tabulated in Table 8.

**Table 8.** The detail features of AlexNet and VGGNet models.

| Attribute | AlexNet | VGGNet |
|-----------|---------|--------|
| Feature layer | conv5 | Conv5-3 |
| Feature map size | $13 \times 13 \times 256$ | $14 \times 14 \times 512$ |
| Receptive field size | $163 \times 163$ | $196 \times 196$ |
| Stride | 16 | 16 |

The NWPU-RESISC45 dataset [48] is used in this subsection. It consists of 31,500 remote sensing images divided into 45 scene classes. Each class includes 700 images with a size of $256 \times 256$ pixels in the red green blue (RGB) color space. This dataset was extracted, by experts in the field of remote sensing image interpretation, from Google Earth (Google Inc.) that maps the Earth by the superimposition of

images obtained from satellite imagery, aerial photography and geographic information system (GIS) onto a 3D globe. This data set is of the largest scale on the number of scene classes and the total number of images. The rich image variations, large within class diversity and high between class similarities make the data set rather challenging. The NWPU-RESISC45 dataset has the following three notable characteristics compared with all existing scene classification datasets including large scale, rich image variation and high with-in class diversity and between class similarity. Figure 5 shows two samples of each class from this dataset.

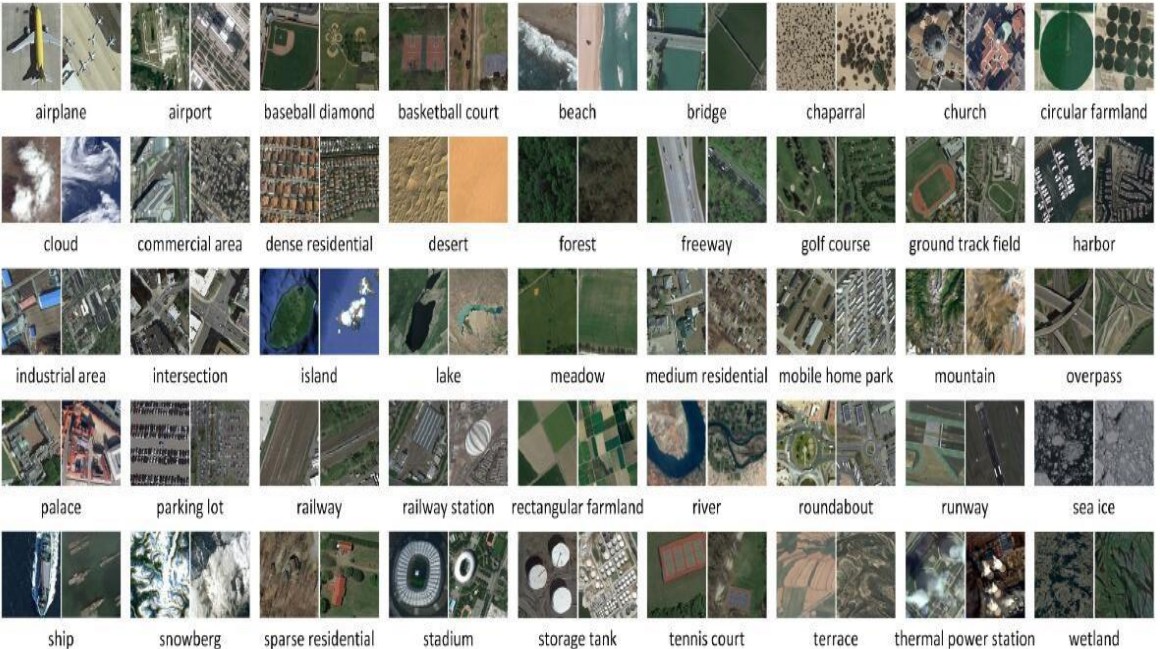

**Figure 5.** Some example images from the NWPU-RESISC45 dataset.

In order to perform a comprehensive comparison, five comparative methods such as Ada-boost (AB), RR, RF, Simple Vote Rule, and QFWEC are used in the experiment. The accuracy results are recorded in Table 9.

**Table 9.** The accuracy results(%) (Average $\pm$ Std) of different methods on deep features.

| Model | Adaboost | RR | RF | Simple Vote Rule | QFWEC | C-LSSVEM |
|---|---|---|---|---|---|---|
| AlexNet | $53.3628 \pm 0.3671$ | $55.6260 \pm 0.6746$ | $55.8264 \pm 0.4926$ | $60.3728 \pm 0.3627$ | $62.5628 \pm 0.8974$ | $\mathbf{67.2631 \pm 0.2345}$ |
| VGGNet | $51.8674 \pm 0.1936$ | $54.4636 \pm 0.4783$ | $54.8946 \pm 0.7836$ | $59.4653 \pm 0.7536$ | $61.6789 \pm 0.984$ | $\mathbf{64.7485 \pm 0.4635}$ |

From Table 9, our proposed C-LSSVEM outperforms all the comparative methods using deep features from AlexNet and VGGNet This indicates the effectiveness of the proposed method on diverse deep features. Our proposed C-LSSVEM achieves higher accuracy and shows better robustness than all the comparatives models. For instance, on AlexNet deep features, Adaboost performed poorly with the least accuracy of mean of 53.3628. RF and RR are similar to a difference of 0.2004. Adaboost again, on the other hand, had the least performance with a mean accuracy of 51.8674. QFWEC performed fairly well on both models. Simple vote rule performed well on AlexNet model compared to VGGNet model. When the proposed C-LSSVEM is applied on AlexNet deep features, the classification accuracy of is 67.2631 and 64.7485 on VGGNet deep features.This indicates that the proposed method has the best classification accuracy.

## 5. Conclusions

In this paper, a novel coupled least squares support vector ensemble machine is presented. We explore the difficulty of how to combine diverse base kernel regressors. Our proposed coupled ensemble model helps to improve the robustness and to produce good classification performance than the single model approach. The coupled least squares support vector ensemble model has the ability to select appropriate kernel types and their parameters in a good coupling strategy with a set of classifiers. We form an ensemble regressor by co-optimizing and weighing base kernel regressors. Experiments conducted on several datasets including artificial datasets, UCI classification datasets, UCI regression datasets, handwritten digits datasets and NWPU-RESISC45 datasets, indicate that C-LSSVEM performs better in achieving minimal regression loss with best classification accuracy relative to selected state-of-the-art regression and classification techniques.

We will aim to expand our model by altering our objective functions into different functions, such as $\varepsilon$-insensitive loss and the hinge loss function in the future. Meanwhile, we will try to find other ways to update the weights of base kernel regressors. Furthermore, we will find more effective ways to utilize the end-to-end deep learning model.

**Author Contributions:** X.-J.S. and D.K.W. fabricated the algorithm; D.K.W. performed the experiments; X.-J.S. analyzed the results and provide supervision; D.K.W. drafted the manuscript; and X.-J.S. reviewed the paper.

**Funding:** This work was funded in part by the National Natural Science Foundation of China (No. 61572240).

**Acknowledgments:** The authors thank Elias Ocquaye, Abeo Timothy Apasiba, Ernest Ganaa, and Huang Chang Bin for their kind assistance, and also thanks to Rita Keddy for their motivation.

**Conflicts of Interest:** The authors declare no conflict of interest.

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
