# Peer review of "Coupled Least Squares Support Vector Ensemble Machines"

_information, doi:10.3390/info10060195_

Reviewer 1 Report

I have revised the paper and seen that the paper present a novel model called C-LSSVEM for classification. The results show that the new model is better than other benchmark models. In general, the paper is good. However, I have several comments for this paper as below:

(1)  It is not clear to me on how this model is developed. I suggest the authors to add the flow chart and give detail description of this novel model, how it is constructed, how the methods are combined to make a novel model.

(2) As I know, the Kernel functions are integrated in LSSVM, so what it the role of these functions in the novel model.

(3) What are disadvantages and advantages of the novel model compared with others, this should be discussed in detail.

(4) What is limitation of the study carried out in this paper, and how other researches can overcome that limitation.

Author Response

Dear Reviewer,

Kindly find attached a word document that addressed all your comments.

Thank you

Reviewer 2 Report

- p1 (L20): Use "etc" or "and so on" in place of "etcetera"
- p1 (L27): Consistency, i.e. either use "LSSVM" or "LS-SVM"; not interchanging.
- p1 (L31): Rephrase sentence around "outliers sensitivity" as does not read correctly.
- p1 (L32): "draw-backs" --> "drawbacks"
- p2 (L77): "using given a sample" --> "using a given sample"
- p3 (L91): "including" --> "include"
- p3 (L99): "Voila" --> "Viola"
- p4 (L142): Shift the indentation to the left so in line with other kernel sub-headings.
- p4 (L143): "..kernel(Radial.." --> "..kernel (Radial.."
- p4 (L145): Need better definition of the kernel parameters (a, b, c). What is a "free parameter" in this context? Likewise for 'sigma' for the "expected value" in this context. For completion, define x and y.
- p5 (L150): Define "RLS".
- p5 (L154): "difficult" --> "difficulty"
- p5 (L161): Repetition "without regressing result without regression result". Rectify.
- p5 (L163-164): Repetition when defining N and Nt.
- p5 (L172): "base regressors." --> "base regressors,"
- p6 (L187-189): Sentence reads as incomplete. Rephrase.
- p7 (L206): "UCI" should be defined.
- p7 (L227): "respectively" --> "are respectively"
- p7 (L236-7): "picture" --> "visualise". Insert "(Figure 1)." at end of sentence - if this does refer to Figure 1.
- p8 (Eq20): What is 'eta'? Noise?
- p9: Rows in Table 1 should match rows in Table 2 and 3.
- p9: What is the benefit of displaying MSE (Table 2) and MAE (Table 3)? Does not Table 2 suffice?
- p9: Consider displaying the numbers in Tables 2 and 3 to 3 decimal places
- p10 (Fig 2): Boxplots suggest skewness of the results, i.e. not normally distributed, so should report medians in place of means in Table 2, or insert an additional column. The same can be said for Fig 3 and Table 3 (however RR and GBR do appear to be normally distributed).
- p11: Similar comments for Table 5 (i.e. 3 decimal places) and Fig 4 for skewness in boxplots (medians).
- p11 (Table 5): Breast cancer result for C-LSSVEM suggests an accuracy of >100%!
- p15 (L379): "keddy" --> "Keddy"
- p15 (L383): "China(" --> "China ("
- p15 (L389): "PloS one" --> "PLoS One"
- p15-17 (Refs 13, 17, 18, 31, 39, 44, 45, 47): Bold the year.

Need to know what software priogramming applications were used, e.g. Matlab, R, Python, C++/C#, etc..

Author Response

Dear Reviewer,

Kindly find attached a word document that addressed all the comments.

Thank you.

This manuscript is a resubmission of an earlier submission. The following is a list of the peer review reports and author responses from that submission.